# Causal Reasoning through Two Cognition Layers
# for Improving Generalization in Visual Question Answering

**Trang Nguyen**
Tokyo Institute of Technology
nguyen.t.ay@m.titech.ac.jp

**Naoaki Okazaki**
Tokyo Institute of Technology
okazaki@c.titech.ac.jp

## Abstract

Generalization in Visual Question Answering (VQA) requires models to answer questions about images with contexts beyond the training distribution. Existing attempts primarily refine unimodal aspects, overlooking enhancements in multimodal aspects. Besides, diverse interpretations of the input lead to various modes of answer generation, highlighting the role of causal reasoning between *interpreting* and *answering* steps in VQA. Through this lens, we propose Cognitive pathways VQA (CopVQA) improving the multimodal predictions by emphasizing causal reasoning factors. CopVQA first operates a pool of pathways that capture diverse causal reasoning flows through *interpreting* and *answering* stages. Mirroring human cognition, we decompose the responsibility of *each* stage into distinct experts and a cognition-enabled component (CC). The two CCs strategically execute one expert for each stage at a time. Finally, we prioritize answer predictions governed by pathways involving both CCs while disregarding answers produced by either CC, thereby emphasizing causal reasoning and supporting generalization. Our experiments on real-life and medical data consistently verify that CopVQA improves VQA performance and generalization across baselines and domains. Notably, CopVQA achieves a new state-of-the-art (SOTA) on the PathVQA dataset and comparable accuracy to the current SOTA on VQA-CPv2, VQAv2, and VQA-RAD, with one-fourth of the model size.

## 1 Introduction

The Visual Question Answering (VQA) task involves answering questions about images, requiring multimodal processing and common sense understanding [Antol et al., 2015]. VQA research has various applications, including autonomous systems [Deruyttere et al., 2019, Zablocki et al., 2022], healthcare [Binh D. Nguyen, 2019, Kovaleva et al., 2020], and education [He et al., 2017].

However, real-life multimodal data diversity poses a challenge for VQA models to achieve Out-of-Distribution (OOD) generalization, which involves performing well on data beyond the training distribution instead of relying on independent and identically distributed (iid) data [Zhang et al., 2021, Goyal and Bengio, 2022, Kawaguchi et al., 2022]. Recent studies have highlighted a risk of OOD generalization in VQA, where models may respond solely based on the question and ignore the input image due to correlations between the question and answer distribution [Niu et al., 2021, Wen et al., 2021] that exist in human knowledge. For example, questions starting with *"Is this...?"* are typically expected to be yes/no questions (*e.g.* , *"Is this a cat?"*) rather than multiple-choice questions (*e.g.* , *"Is this a cat or dog?"*), leading to possible correct answers with a simple *"yes"* or *"no"*.

There are many attempts to solve this issue, such as (1) reducing the linguistic correlation [Niu et al., 2021, Wen et al., 2021] by avoiding answers generated from only the question used as input, (2) strengthening the visual processing [Yang et al., 2020], and (3) balancing the answer distribution by generating new image-question pairs [Chen et al., 2020, Gokhale et al., 2020, Si et al., 2022]. However, these approaches tend to overlook the critical aspect of enhancing multimodal predictions, instead focusing on unimodal aspects (either language or visual) or the data itself. Our assumption is that enhancing the quality of multimodal predictions would be a potential route for improving generalization in VQA.

In fact, solving the VQA task requires the integration of multimodal processing and common sense knowledge. Consequently, VQA can be conceptualized as a two-stage process: *input interpreting* and *answering*, which involves generating an interpretation of the multimodal input and answering the question by querying the knowledge space. Besides, similarly to how humans tackle the

VQA task, the input misunderstanding can harm the answering stage, posing a risk to the overall performance of VQA. Therefore, comprehending the causal reasoning behind this two-stage process becomes crucial to improving the VQA task.

In this work, we propose Cognitive pathways VQA (CopVQA) to boost the causal reasoning in VQA to enhance the OOD generalization. CopVQA derives from the findings of *knowledge modularity* and *cognitive pathways* from cognitive neuroscience [Baars, 2005, Kahneman, 2011, Goyal and Bengio, 2022], which mentions (1) the human brain organizes independent modules to handle distinct knowledge pieces and (2) the communication efficacy among these modules supports the generalization. We decompose each *interpreting* and *answering* stage into a set of experts and a cognition-enabled component (CC) strategically activates one expert for each stage at a time. By this approach, CopVQA disentangles the VQA task into specialized experts connected by pathways through *interpreting-answering* process. Subsequently, we extend the biases in VQA that are also from the monolithic procedure (instead of expert selection) besides the linguistic correlation. Finally, we emphasize answers governed by the disentangled stages with multimodal input and disregard other answers, including ones from the monolithic procedure and from unimodal input.

The contributions of this work are summarized as follows: (1) we propose CopVQA to improve OOD generalization by enhancing causal reasoning that is compatible with diverse VQA baselines and domains; (2) to our best knowledge, CopVQA is the first work that formulates VQA as two layers of cognitive pathways, further facilitating research on causal reasoning in VQA; and (3) we achieve the new SOTA for the PathVQA dataset and mark the comparable results to the current SOTAs of VQA-CPv2, VQAv2, and VQA-RAD datasets with only one-four of the model sizes.

## 2 Related Work

The generalization restriction in VQA arises from biases between questions and answers in human knowledge, where certain question types exhibit strong correlations with predictable answers based on common knowledge. These biases are also reflected in VQA datasets. For example, in the VQAv1 dataset [Antol et al., 2015], a VQA model can quickly achieve an accuracy of around 40% on sport-related questions by simply providing the answer "tennis." To further challenge the generalization ability of the VQA models, the VQA Changing Priors (VQA-CPv2) dataset [Agrawal et al., 2017] is introduced. This dataset is deliberately designed to feature different answer distributions between its train and test sets. Therefore, the emergence of the VQA-CP dataset has brought about a significant shift in the VQA landscape, demanding that VQA models go beyond exploiting correlations and overcoming biases from the given training data.

Among attempts of capturing language biases, RUBi [Cadene et al., 2019] introduces a noteworthy approach that computes the answer probability using only the input question, thereby capturing linguistic biases. Then, the question-only branch carrying the biases is used to compute a mask that aims to mitigate biased prediction of unimodal input from the multimodal prediction. Building upon this line of research, CFVQA [Niu et al., 2021] pioneers a comprehensive causal-effect view of VQA that subtracts the impact of the question-only branch, represented as the answer logit, from the overall answer logit (further discussed in Section 3). Similarly, DVQA [Wen et al., 2021] proposes subtracting the question-only branch's impact from the multimodal branch in the hidden layer. By doing so, DVQA aims to mitigate biased predictions before passing the output through the classifier model for final answer prediction.

In this study, we delve deeper into capturing biases using separate branches and subsequently eliminate them. However, besides the linguistic correlations, we assume another potential restriction in VQA generalization is the monolithic approach across the diverse scenarios of the multimodal input. Therefore, we attempt to eliminate answers by monolithic procedures involved in *interpreting* and *answering* stages parallelly emphasize answers by disentanglement approach.

## 3 Preliminaries: Causality view in VQA

Introduced in CFVQA [Niu et al., 2021], the causal graph of VQA is depicted in Figure 1a in which the inputs $V$ and $Q$ cause an answer $A$, and a mediator[1] $K$ represents the knowledge space. We have the **direct paths**, which are $Q \rightarrow A$ and $V \rightarrow A$, representing the answers based solely on unimodal

---

[1]To explore the reasons for input effecting on the output, Pearl [2009], Pearl and Mackenzie [2018] mention the term *mediator* to dissect the effect into direct and indirect effects.

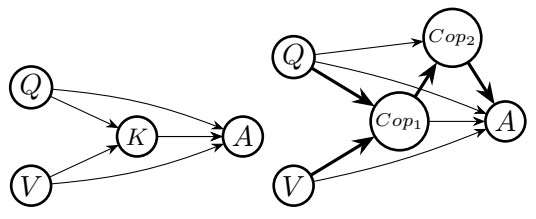

(a) Conventional VQA    (b) Cognitive pathways VQA

Figure 1: The conventional causal-effect view of the VQA task and the proposed Cognitive pathways VQA, which enhances the causal reasoning aspect

input (either question or image). In contrast, the **indirect path**, $(V, Q) \rightarrow K \rightarrow A$, represents the answer by considering the multimodal input and the interaction in the knowledge space.

With a single mediator $K$, the idea of CFVQA is to capture the language prior by the $Q \rightarrow A$ branch, which involves only the question and produces an answer logit $Z_q$. Likewise, the answer produced through $K$ with multimodal input is $Z_k$. Subsequently, they produce the final answer logit as follows, where $c$ is a parameter for normalization:

$$Z_{final} = \log\sigma(Z_k + Z_q) - \log\sigma(Z_q + c)$$

Regarding optimization, they utilize Cross-Entropy loss (CELoss) and define $\mathcal{L} = \text{CELoss}(\log\sigma(Z_k + Z_q), a) + \text{CELoss}(\log\sigma(Z_q), a)$.

# 4 Cognitive pathways VQA - CopVQA

In this section, we present CopVQA from multiple viewpoints, which are the overview in Section 4.1, the causal-effect view of CopVQA in Section 4.2, and the CopVQA implementation in Section 4.3.

## 4.1 CopVQA overview

The CopVQA, depicted in Figure 2, serves as a VQA backbone that emphasizes causal reasoning in multimodal prediction. CopVQA mitigates the negative consequences of disregarding causal reasoning, yet enhancing generalization. Initially, we explore the knowledge space as **(1)** multimodal knowledge for *interpreting* the multimodal input, and **(2)** commonsense knowledge for *answering* based on the interpretation obtained. Additionally, assuming that different interpretations of the input lead to diverse ways of answering, we perceive the use of monolithic procedures for *interpreting* and *answering* across diverse multimodal input as a potential bias that hampers generalization, extending beyond the linguistic correlations discussed in prior

work. Consequently, we define a non-biased approach as one that commits to integrating the multimodal input and strategically considering proper knowledge pieces for *interpreting* and *answering*, rather than relying on monolithic procedures.

Each *intepreting* and *answering* stages involves diverse distinct experts with a cognition-enabled component (CC) that activates one expert at a time. Consequently, we define a complete reasoning flow as selecting an appropriate expert pair for the *interpreting-answering* process. In contrast, incomplete reasoning flows rely on monolithic procedures or utilize unimodal input. Finally, we attempt to emphasize the prediction obtained through the full reasoning flow and disregard the ones from incompleted reasoning flows.

## 4.2 CopVQA from causal-effect view

To establish a solid connection between the comprehensive overview and implementation details of CopVQA, we present the causal-effect view of the proposed architecture in this section.

The causal view of CopVQA presented in Figure 1b, contains the input pair $(V, Q)$ that leads to an answer $A$, controlled by the two sets of cognitive pathways denoted as mediators $Cop_1$ and $Cop_2$. Specifically, we have **direct paths**: $Q \rightarrow A$ and $V \rightarrow A$; and **indirect paths** including **Case 1**: $(V, Q) \rightarrow Cop_1 \rightarrow A$, **Case 2**: $Q \rightarrow Cop_2 \rightarrow A$, and **Case 3**: $(V, Q) \rightarrow Cop_1 \rightarrow Cop_2 \rightarrow A$.

Specifically, any path that does not involve $Cop_1$ ($Q \rightarrow A$, $V \rightarrow A$, and $Q \rightarrow Cop_2 \rightarrow A$) are categorized as *unimodal* paths, as it does not involve multimodal interpretation. Likewise, indirect paths that bypass $Cop_2$ (*e.g.* $(V, Q) \rightarrow Cop_1 \rightarrow A$) are considered *monolithic* ones[2], as it does not involve expert selection for both *interpreting* and *answering*. Finally, CopVQA emphasizes the effect from completed reasoning flow, which is $(V, Q) \rightarrow Cop_1 \rightarrow Cop_2 \rightarrow A$, and eliminates effects of incompleted reasoning flows, including *unimodal* and *monolithic* paths.

## 4.3 Implementation Details

We design cognitive pathways as a Mixture of Experts [Jacobs et al., 1991] (MoE), which disentangles a task into specialized experts. Mathematically, an MoE setup $\mathcal{M}$ contains **(1)** $N$ experts $\{E_1, E_2, \ldots, E_N\}$ with distinct parameters and **(2)**

---

[2]$Q \rightarrow Cop_2 \rightarrow A$ is also a *monolithic* path as it does not involve both cognition layers. However, for simplicity, we only mention this path as an *unimodal* path.

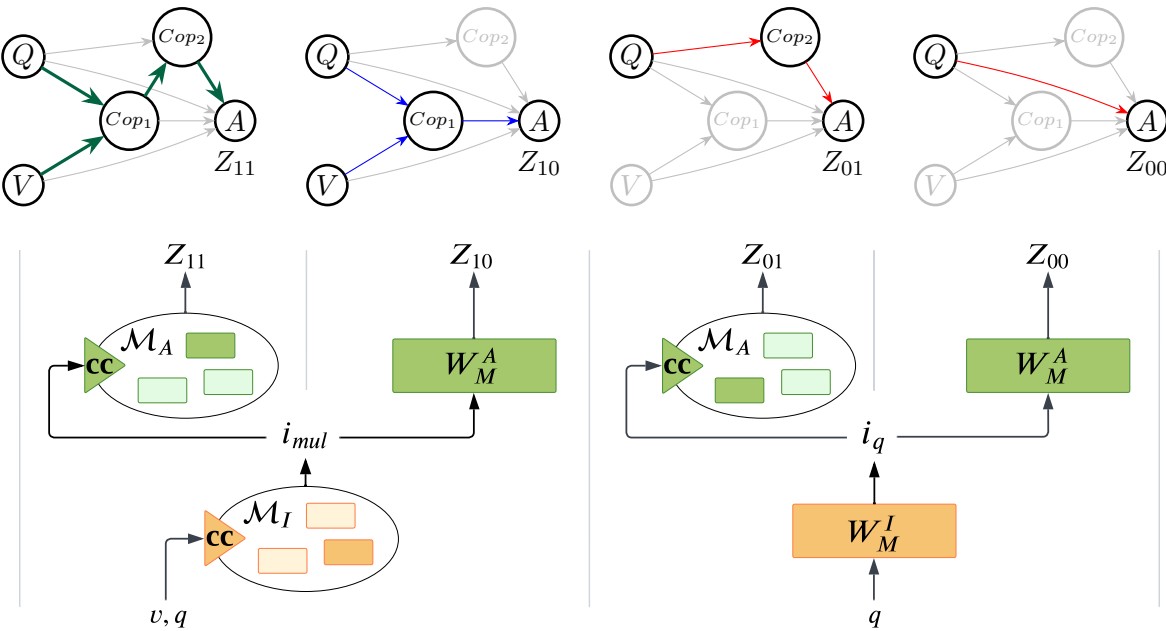

(a) The causal graphs of four different kinds of reasoning flows and the corresponding computational flows in CopVQA

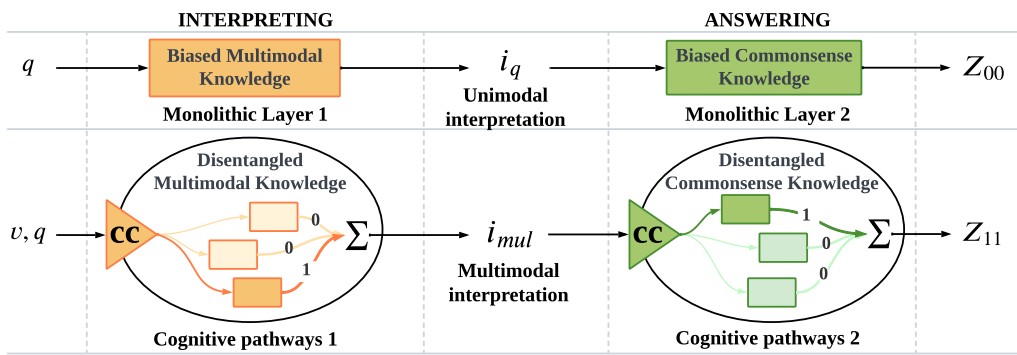

(b) Visualization of *interpreting-answering* processes through monolithic and disentangled knowledge spaces in computing $Z_{00}$ and $Z_{11}$.

Figure 2: CopVQA from the causal-effect view aligned to computational flows. CopVQA emphasizes the full causal reasoning flow (green), simultaneously eliminating effects from *monolithic* (blue) and *unimodal* (red) flows.

gating model $\mathcal{G} : \mathbb{R}^d \to \mathbb{R}^N$. As described in Equation 1, with input $x \in \mathbb{R}^d$, the output $y$ is the sum of results from $N$ experts weighted by $g$, produced by $\mathcal{G}$. We conduct $g = \text{Gumbel-max}(\mathcal{G}(x))$ with a Gumbel-max to achieve a *1-hot-like* probability.

$$y = \sum_{n=1}^{N} g_i \times E_n(x), \qquad (1)$$

### 4.3.1 Computational Flow in CopVQA

We describe the computational flow in CopVQA by introducing the notations and 2-stage process.

Given the pre-processed question $q \in \mathbb{R}^{d_q}$ and image $v \in \mathbb{R}^{d_v}$, where $d_q$ and $d_v$ are the shapes of the modalities, the VQA model aims to predict the answer $\hat{a}$, which is an index in the vocabulary set size of $V$. In the *interpreting* stage, CopVQA

produces an interpretation as a $d_i$-dimensional vector, denoted as $i \in \mathbb{R}^{d_i}$. In the *answering* stage, CopVQA conducts a classifier model that outputs $Z_{jk} \in \mathbb{R}^V$, where $j, k \in \{0, 1\}$. Specifically, $j$ is 1 when the interpretation $i$ is governed by experts from $Cop_1$ in *interpreting*, and 0 otherwise (governed by a monolithic procedure). We analogously define $k \in \{0, 1\}$ for $Cop_2$ in the *answering* stage. Obtaining the answer $\hat{a}$ can be done by getting the max value's index of $\log \sigma(Z_{jk})$.

**Notation 1**: Denote $\mathcal{M}_I$ and $\mathcal{M}_A$ as MoE setups align $Cop_1$ for *Interpreting* and $Cop_2$ for *Answering*, respectively. $N_1$ experts in $\mathcal{M}_I$ is designed to map $\mathbb{R}^{d_q} \to \mathbb{R}^{d_i}$ responding to the *interpreting* stage; similarly, $N_2$ experts in $\mathcal{M}_A$ map $\mathbb{R}^{d_i} \to \mathbb{R}^V$ responding to the *answering* stage. The cognition-enabled components align to $\mathcal{G}(\cdot)$

in each MoE, which are $\mathcal{G}_I : \mathbb{R}^{d_q} \to \mathbb{R}^{N_1}$ and $\mathcal{G}_A : \mathbb{R}^{d_i} \to \mathbb{R}^{N_2}$ for the *interpreting* and *answering* stages, respectively.

**Notation 2**: We denote monolithic procedures $W_M^1 : \mathbb{R}^{d_q} \to \mathbb{R}^{d_i}$ and $W_M^2 : \mathbb{R}^{d_i} \to \mathbb{R}^V$ for the *interpreting* and *answering* stages, respectively.

**Stage 1 - Input interpreting**: Let $i_{mul} \in \mathbb{R}^{d_i}$ be the multimodal interpretation and $i_q \in \mathbb{R}^{d_i}$ be the unimodal interpretation, described in Equation 2. Specifically, $i_{mul}$ is computed by passing $q$ through $Cop_1$ and subsequently applying a Fusion function (follow the baselines, details in Section 5.3) on $v$ to obtain a multimodal interpretation. Likewise, $i_q$ is obtained from the input question $q$ alone by $W_M^1$.

$$i_{mul} = \mathsf{Fusion}(Cop_1(q), v), \quad i_q = W_M^1(q) \quad (2)$$

**Stage 2 - Answering**: In this step, we compute the pool of outputs from multiple reasoning flows. Let $Z_{11}$, described in Equation 3, align to the full causal reasoning flow governed by both cognition layers. Likewise, $Z_{10}$, as in Equations 4, represents the monolithic path that involves only $Cop_1$. Finally, $Z_{01}$ and $Z_{00}$, formulated in Equations 5 and 6, represent for the output from unimodal paths.

$$Z_{11} = Cop_2(i_{mul}) \quad (3)$$
$$Z_{10} = W_M^1(i_{mul}) \quad (4)$$
$$Z_{01} = Cop_2(i_q) \quad (5)$$
$$Z_{00} = W_M^2(i_q) \quad (6)$$

#### 4.3.2 Training and Inference Time

**Output finalizing** As discussed in Section 4.2, CopVQA emphasizes the impact of the fully causal reasoning flow and eliminates the impacts of the incompleted reasoning flows. Inspired by the Niu et al. [2021] as introduced in Section 3, we design strategies for output finalizing, loss functions, and answer finalizing for the inference time in Equations 7, 8, and 9, respectively, with $a$ is the target answer and the LossFn function is inherited from particular baselines.

$$Z_{final} = \log\sigma(Z_{11} + Z_{10} + Z_{01} + Z_{00}) \\ - \log\sigma(Z_{10} + Z_{01} + Z_{00}) \quad (7)$$

$$\mathcal{L}_{total} = \mathsf{LossFn}(\log\sigma(Z_{11}+Z_{10}+Z_{01}+Z_{00}), a) \\ \mathcal{L}_M = \mathsf{LossFn}(\log\sigma(Z_{10}), a) \\ \mathcal{L}_U = \mathsf{LossFn}(\log\sigma(Z_{01}), a) \\ + \mathsf{LossFn}(\log\sigma(Z_{00}), a) \\ \mathcal{L}_{CopVQA} = \mathcal{L}_{total} + \mathcal{L}_M + \mathcal{L}_U \quad (8)$$

$$\text{Inference: } \hat{a} = \mathsf{argmax}(\log\sigma(Z_{final})) \quad (9)$$

**Model architecture** Experts in the Model of Experts (MoE) framework have a common architecture but carry distinct parameters. Denote a layer $l : \mathbb{R}^{in} \to \mathbb{R}^{out}$ with a hidden size $h$ in a particular baseline that is in charge of *interpreting* or *answering* stages, we design $l' : \mathbb{R}^{in} \to \mathbb{R}^{out}$ with the hidden size $h'$. Subsequently, individual experts and the monolithic models in CopVQA share the same architecture as $l'$. Precisely, $h'$ is fine-tuned to achieve the optimal result. Through experimentation, we have discovered that selecting values for $h'$ such that the total number of parameters in all experts and monolithic models does not exceed the number of parameters in $l$ tends to yield optimal results. This observation regarding the adjustment of $h'$ aligns with the principles of knowledge modularity, which we discuss in more detail in Appendix C. As a result, CopVQA does not increase parameters beyond those present in the baseline model.

## 5 Experiment Setup

We conducted experiments to validate: $\mathcal{H}_1$ - Causal reasoning in multimodal prediction benefits VQA performance (**Section** 6.1), $\mathcal{H}_2$ - Causal reasoning in multimodal prediction enhances OOD generalization (**Section** 6.1), and $\mathcal{H}_3$ - The disentangled architecture is crucial for reasoning (**Section** 6.2).

### 5.1 Datasets

To examine $\mathcal{H}_1$, we conduct experiments on four datasets in two domains: (1) real-life images: **VQA-CPv2** [Agrawal et al., 2017] and **VQAv2** [Goyal et al., 2017] and (2) medical data: **PathVQA** [He et al., 2021] and **VQA-RAD** [Lau et al., 2018]. Questions in VQA-CPv2 and VQAv2 are divided into three types: "Yes/No" (Y/N), Number (Num.), and Other, with 65 categories of question pre-fix (such as "Is it...?"). PathVQA categorizes "Y/N" questions separately from "Free-form" questions, whereas VQA-RAD categorizes questions with limited answer candidates as "Close" and the remaining questions as "Open" type.

To examine $\mathcal{H}_2$, we investigate results on the VQA-CPv2 dataset, a valuable benchmark for assessing OOD generalization in VQA. This dataset features substantial variations in answer distribution per question category between the training and test sets, making it an ideal choice for evaluating the models' ability in OOD scenarios.

| Test set | VQAv2 | | | | | VQA-CPv2 | | | | |
| Method | Y/N | Num. | Other | Overall | Gap | Y/N | Num. | Other | Overall | Gap |
|---|---|---|---|---|---|---|---|---|---|---|
| RUBi[†] | - | - | - | 61.1 | - | 68.7 | 20.3 | 43.2 | 47.1 | - |
| SCR[†] | 78.8 | 41.6 | 54.5 | 62.2 | - | 72.4 | 10.9 | 48.0 | 49.5 | - |
| Mutant[†] | 82.1 | 42.5 | 53.3 | 62.6 | - | 88.9 | 49.7 | 50.7 | 61.7 | - |
| CFVQA | $81.3^{\pm0.2}$ | $43.4^{\pm0.3}$ | $50.1^{\pm0.1}$ | $60.7^{\pm0.1}$ | - | $90.4^{\pm0.3}$ | $21.3^{\pm0.7}$ | $45.2^{\pm0.2}$ | $55.0^{\pm0.2}$ | - |
| +CopVQA | $\mathbf{81.4^{\pm0.3}}$ | $\mathbf{43.8^{\pm0.2}}$ | $\mathbf{52.4^{\pm0.2}}$ | $\mathbf{62.2^{\pm0.3}}$ | +1.5 | $\mathbf{91.1^{\pm0.3}}$ | $\mathbf{41.6^{\pm0.2}}$ | $\mathbf{46.4^{\pm0.1}}$ | $\mathbf{57.8^{\pm0.3}}$ | +2.8 |
| DVQA | $81.7^{\pm0.3}$ | $42.8^{\pm0.4}$ | $56.7^{\pm0.2}$ | $64.3^{\pm0.3}$ | - | $88.5^{\pm0.2}$ | $48.7^{\pm0.6}$ | $50.1^{\pm0.1}$ | $61.1^{\pm0.1}$ | - |
| +CopVQA | $\mathbf{82.6^{\pm0.2}}$ | $\mathbf{45.2^{\pm0.3}}$ | $\mathbf{59.0^{\pm0.3}}$ | $\mathbf{67.5^{\pm0.2}}$ | +3.2 | $\mathbf{92.1^{\pm0.3}}$ | $\mathbf{59.4^{\pm0.4}}$ | $\mathbf{61.4^{\pm0.3}}$ | $\mathbf{67.9^{\pm0.3}}$ | +6.8 |

Table 1: Accuracy comparison on VQAv2 and VQA-CPv2 datasets. The best scores are bolded.

| Test set | PathVQA | | | | VQA-RAD | | | |
| Method | Y/N | Free-form | Overall | Gap | Open | Close | Overall | Gap |
|---|---|---|---|---|---|---|---|---|
| MMQ | $83.6^{\pm0.4}$ | $13.5^{\pm0.5}$ | $48.6^{\pm0.2}$ | - | $52.4^{\pm1.4}$ | $75.3^{\pm1.1}$ | $66.8^{\pm0.4}$ | - |
| + CopVQA | $\mathbf{85.3^{\pm0.1}}$ | $\mathbf{16.6^{\pm0.1}}$ | $\mathbf{50.9^{\pm0.3}}$ | +2.3 | $\mathbf{56.5^{\pm0.9}}$ | $\mathbf{77.1^{\pm0.6}}$ | $\mathbf{70.2^{\pm0.3}}$ | +3.4 |

Table 2: Accuracy comparison on PathVQA and VQA-RAD datasets. The best scores are bolded.

## 5.2 Baselines

We implement and compare CopVQA to baselines that do not emphasize the causal reasoning in multimodal prediction. In VQA-CPv2 and VQAv2, we implement CopVQA on **CFVQA** [Niu et al., 2021] and **DVQA** [Wen et al., 2021] baselines that attempts to eliminate the language priors. Besides, to fulfill the comparison, we compare CopVQA to approaches that strengthen visual processing: **SCR** [Wu and Mooney, 2019]; balancing answers distribution: **Mutant** [Gokhale et al., 2020][3]. In medical datasets, we conduct CopVQA on PathVQA's SOTA - **MMQ** [Binh D. Nguyen, 2019], which leverage significant features by meta-annotation.

## 5.3 Implementation details

We fine-tune $N_1$ and $N_2$ with each baseline and maintain consistent configurations, including the optimizer, number of epochs, and batch sizes. For the DVQA baseline, the best setting for the $(N_1, N_2)$ pair is $(3, 3)$. In contrast, for the CFVQA and MMQ baselines, the optimal pair is $(5, 5)$.

CFVQA employs Fusion as the *Block factory fusion* from Ben-Younes et al. [2019] and trains by CrossEntropy loss. Likewise, based on the UpDn [Anderson et al., 2017], DVQA designs Fusion as the multiplication of the pre-processed $v$ and $q$ and uses BinaryCrossEntropy as LossFn. On the other hand, the baseline MMQ, based on BAN [Kim et al., 2018], designs Fusion as the recurrent multiplication of the processed $v$, $q$, and BAN's result on $v$ and $q$, and utilizes the BinaryCrossEntropy loss.

---

[3]We compare to Mutant that based on UpDn [Anderson et al., 2017] since DVQA and our DVQA-based CopVQA share the common foundation of UpDn.

## 6 Experimental Results and Discussion

### 6.1 Quantitative Results

Overall, CopVQA outperforms all baselines in both iid and OOD. Specifically, Table 1 presents results on VQAv2 and VQA-CPv2, and Table 2 shows results on PathVQA and VQA-RAD. Baselines marked with a "†" are reported results from the original paper, while others are our reproduced mean and standard error on 5 random seeds. The "Gap" shows the improvement of CopVQA from the reproduced baseline, with a "-" for unreported results. In addition, Table 4 in the Appendix compares the sizes and training time between the models.

Proving $\mathcal{H}_1$, CopVQA reveals the consistent improvement in all answer categories regardless of baselines and domains. Notably, in VQAv2, CopVQA achieves a +1.5% and +3.2% higher than CFVQA and DVQA baselines, respectively, in the *Overall* score. In medical datasets, CopVQA marks the improved ability in the medical VQA task with a +2.3 and +3.4 points higher MMQ baseline.

Proving $\mathcal{H}_2$, regardless of the based models, CopVQA demonstrates the effectiveness of causal reasoning in generalization by outperforming baselines in VQA-CPv2 by a large margin. Specifically, CopVQA acquires a remarkable improvement of +2.8% in CFVQA-based and +6.8% in DVQA-based in the *Overall* score. Notably, CopVQA shows exceptional performance on the *Number* type, with +20.3 and +12.7 points increased from the CFVQA and DVQA baselines, respectively.

Compared to current SOTAs, CopVQA achieves a new SOTA on PathVQA with +2.3 points higher than **MMQ**, the previous SOTA. Besides, CopVQA

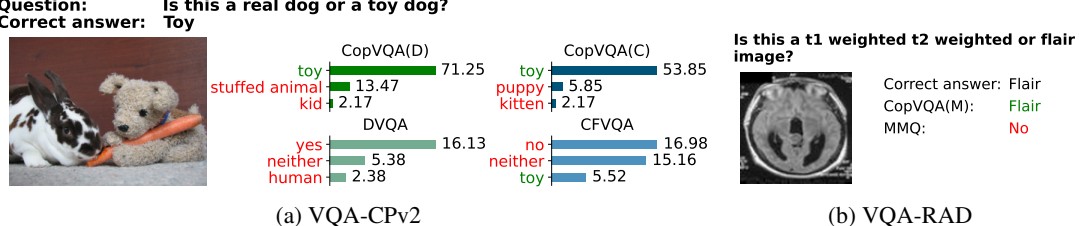

(a) VQA-CPv2                                    (b) VQA-RAD

Figure 3: Sample of debiased cases, listing the labels with top probability. The green and red labels are the correct and incorrect answers, respectively.

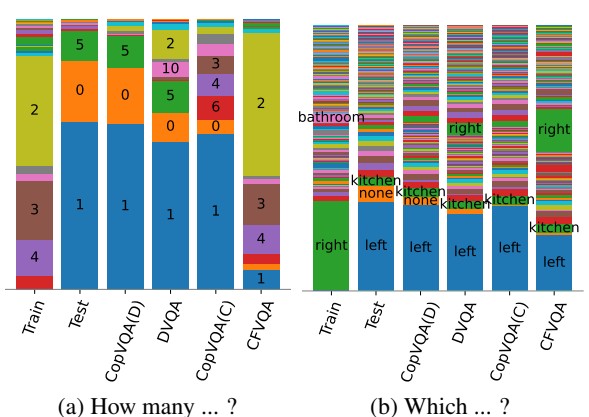

(a) How many ... ?                (b) Which ... ?

Figure 4: Answer distributions on VQA-CPv2 and the comparison of debiasing ability. CopVQA produces the most similar distributions as in the test set.

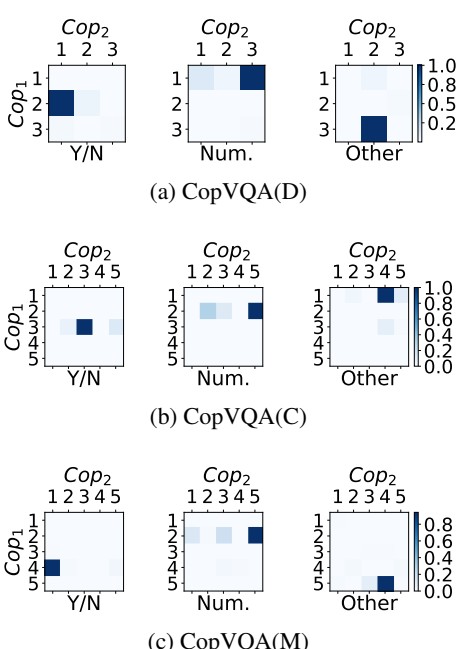

(a) CopVQA(D)

(b) CopVQA(C)

(c) CopVQA(M)

Figure 5: The proportion of experts selection from $Cop_1$ and $Cop_2$ assignment during the inference time.

is comparable to MMBS [Si et al., 2022], the current SOTA of VQA-CPv2 and VQAv2, with only one-fourth of the model size (details in Table 4 in Appendix). Specifically, MMBS gains 68.39% and 69.43% *Overall* scores on VQA-CPv2 and VQAv2, respectively, with over 200M parameters, while DVQA+CopVQA marks 67.9% and 67.5% with less than 52M parameters. Lastly, CopVQA marks 1.9 points lower than CLIP [Eslami et al., 2021] - current VQA-RAD's SOTA.

Compared to other approaches in VQA-CPv2 and VQAv2, Table 1 indicates that CopVQA is significantly better than SCR. Specifically, CopVQA achieves a comparable accuracy or even higher than Mutant without data augmentation.

## 6.2 Qualitative Results

We investigate $\mathcal{H}_3$ by the underlying performance of CopVQA on VQA-CPv2 and VQA-RAD. We denote CopVQA based on DVQA and CFVQA on VQA-CPv2 as CopVQA(D) and CopVQA(C), and based on MMQ on VQA-RAD as CopVQA(M).

**Discussion on the debiased samples** Figure 3 compares debiasing results with values obtained

by applying softmax on the answer logit. In Figure 3a shows that CopVQA accurately answers the question by treating it as a multiple-choice type, whereas the baselines are trapped by bias toward Yes/No type, resulting in incorrect answers. In addition, CopVQA models list other high-probability answers relevant to the correct one, such as "stuffed animal" or "puppy". In Figure 3b, CopVQA gives the correct answer, while MMQ struggles to recognize the question's purpose and gives an incorrect one. We provide more samples in Appendix B.

**Debiased answers distribution** Figure 4 depicts the distribution of answers collected from the entire train set, test set, and models' prediction during inference. The analysis demonstrates that CopVQA(D) produces the most accurate distribution that resembles the test set by mitigating the biased answers found in DVQA in both examples. Likewise, CopVQA(C) exhibits the generalizabil-

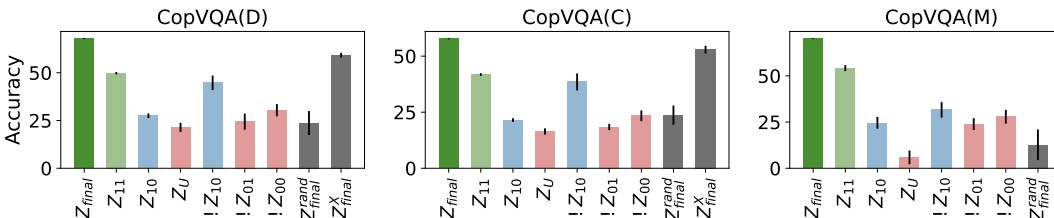

Figure 6: Ablation studies to clarify the individual effect of components in CopVQA design. The original design CopVQA demonstrates the dominant performance over all variants.

ity over CFVQA as it removes the strongly biased answers "2" in Figure 4a and "right" in Figure 4b. Appendix B discusses more on models' success and failure cases in the debiased answer distribution.

**Discussion on mechanisms selection** Figure 5 presents the proportion of expert-pair selection from $Cop_1$ and $Cop_2$ over $N_1 \times N_2$ combinations. CopVQA consistently assigns the non-overlapping of experts to separate answer categories. For instance, in CopVQA(D), $Cop_1$ assigns experts 2, 1, and 3 for the *Y/N*, *Number*, and *Other* types, respectively, in most of cases. This pattern aligns with the strategy of CopVQA, where experts specialize in distinct contexts of the multimodal input and commonsense knowledge space. Moreover, we observe that CopVQA allot more than one *answering* expert for *Number* type by a visible proportion, which adapts to the diverse skills required, explaining the significant improvement of CopVQA on this type.

## 7 Ablation Studies

**Individual effects in CopVQA** We conduct ablations to observe the effect of components in Equations 7. The analysis confirms the dominance of CopVQA over the modified versions. Figure 6 compares the *Overvall* score of $Z_{final}$ to ones from:
(1) $Z_{11}$, $Z_{10}$, and $Z_U$ (the sum of $Z_{01}$ and $Z_{00}$). Values of $Z_{10}$ are significantly lower than those of $Z_{final}$, supporting the assumption that disentangled architectures benefit VQA.
(2) $!Z_{11}$, $!Z_{10}$ and $!Z_{00}$ denote results of $Z_{final}$ when $Z_{11}$, $Z_{10}$, or $Z_U$ is excluded from $Z_{final}$ in inference, respectively. It indicates the importance of *monolithic* and *unimodal* paths when $!Z_{10}$, $!Z_{01}$, and $!Z_{00}$ are far from $Z_{final}$.
(3) $Z_{final}^{rand}$ as the score of $Z_{final}$ when experts are randomly selected in inference. $Z_{final}^{rand}$ drops dramatically, proving the vital role of selecting a proper pair of experts to improve accuracy.
(4) $Z_{final}^X$ as the score of $Z_{final}$ when we break the causal structure in Figure 2 and incorpo-

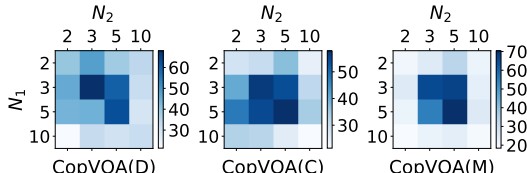

Figure 7: *Overall* accuracy comparison over various configurations of $N_1$ and $N_2$ in CopVQA.

rate $v$ in computing $Z_{01}$, or designing $Z_{01}^X = Cop_2(\text{Fusion}(W_M^{1,X}(q), v))$. $Z_{final}^X$ is comparable but does not exceed $Z_{final}$ in all models, proving the pivot of the original causal structure design and the balance of *monolithic* and *unimodal* paths.
We collect scores from the best checkpoint for CopVQA(M) and $Z_{final}^{rand}$, and from the best score over epochs for CopVQA(D) and CopVQA(C).

**Number of experts in** $Cop_1$ **and** $Cop_2$ Figure 7 indicates results from multiple $(N_1, N_2)$ pairs. We observe that values in the range of 3 to 5 experts tend to yield higher scores, with the pair of duplicated values achieving the highest scores, while pairs with values of 2 and 10 result in significantly lower scores. This finding suggests that an appropriate balance between experts in two sets of pathways is crucial to acquire high performance.

## 8 Conclusion

We proposed CopVQA, a novel framework to improve OOD generalization by leveraging causal reasoning in VQA. CopVQA emphasizes the answer with a full reasoning flow governed by disentangled knowledge space and cognition-enabled component in both *interpreting* and *answering* stages while eliminating answers in incompleted reasoning flows, which involve unimodal input or monolithic procedures. CopVQA outperforms baselines across domains in iid and OOD. Notably, CopVQA achieves new SOTA on PathVQA and comparable results with the current SOTAs of other datasets with significantly fewer parameters.

## Limitations

Despite being touted as a robust backbone that enhances generalization in the VQA task through its emphasis on causal reasoning in multimodal processing, the CopVQA method exhibits certain limitations that should be acknowledged, including:

- **Sensitive in data with limited occurrences** such as brand names or country names, which poses challenges for effective debiasing. We further discuss this point in Appendix B.2.

- **Require careful finetuning** for each baseline and dataset, involving tuning hyperparameters like $N_1$, $N_2$, and experts' architecture to achieve optimal performance.

## Acknowledgement

These research results were partly obtained from the commissioned research (No. 225) by the National Institute of Information and Communications Technology (NICT), Japan.

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

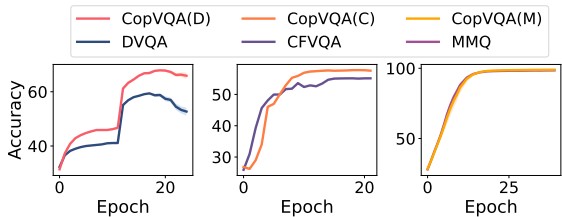

Figure 8: Comparison of validation results during the training process.

## A  Further discussion on experiment details

### A.1  Datasets

The VQA-CPv2 and VQAv2 are two popular datasets in the VQA domain. We follow a standardized downloading process and input preprocess of CFVQA for CopVQA(C) and from DVQA for CopVQA(D). This ensures consistency and comparability across different approaches. Similarly, in the case of PathVQA and VQA-RAD from the MMQ baseline, similar guidelines are followed to ensure a standardized experimental setup. In addition, Table 3 indicates the training, validation, and test splits of each datasets.

### A.2  Validation along epochs

In Figure 8, the validation results indicate that the proposed CopVQA outperforms the baseline models DVQA and CFVQA by a significant margin. The performance of CopVQA is noticeably better, showcasing its effectiveness in visual question answering. Additionally, when compared to the MMQ baseline, CopVQA demonstrates comparable performance, with a slight improvement. These findings highlight the superiority of CopVQA in addressing the task of VQA, surpassing existing baseline models and exhibiting promising potential in the field.

### A.3  The role of disentangling ability

To verify the role of the disentangling ability of experts, we conduct an ablation that adjusts the number of experts to be activated by $\mathcal{G}$ of both Cognitive pathways and in both training and inference time. Particularly, we modify the $\mathcal{G}$ to achieve a *k-hot-like* instead of *1-hot-like* probability.

The comparison in Figure 9 demonstrates the dominant performance of CopVQA with only one expert activated by each set of pathways. Likewise, increasing $k$ to 2 or 3 significantly drops VQA

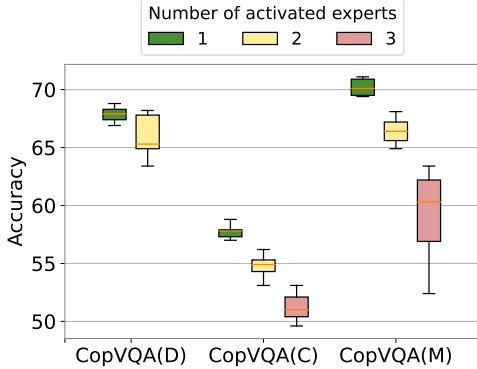

Figure 9: Ablation study on the essential role of the disentangling ability of experts.

accuracy, with a broader range of errors. This analysis results confirm the essential role of the disentangling ability of each expert that learns distinct knowledge to enhance the performance and reserve the robustness of the proposed CopVQA.

## B  Further discussion on qualitative analysis

### B.1  Debiased samples

Figure 10 presents cases where CopVQA correctly answers questions that other baselines fail to answer correctly. The primary reason for this success is the improved understanding of the question's purpose by CopVQA. It effectively addresses biases and avoids potential pitfalls that lead to incorrect answers. Additionally, in VQA-CPv2, CopVQA takes into consideration the relationships between related answers, which further enhances its accuracy and robustness. For instance, the answers "outdoor" and "indoor" have a high probability in the third sample, while baselines consider "yes" and "no" as other potential answers. Overall, the qualitative analysis demonstrates the effectiveness of CopVQA in mitigating biases and improving the performance of VQA systems.

Figure 11 shows cases where all models' predictions are correct. It is observed that in the majority of these cases, the proposed method CopVQA generates a higher probability for the correct answer compared to baselines. The higher probabilities assigned by CopVQA reflect its ability to capture important visual cues, contextual information, and semantic relationships in disentangled architectures, thereby outperforming the baseline models in terms of answer quality. This analysis highlights the superior performance of CopVQA in producing reliable and accurate answers in VQA.

| Datasets | Num. Questions | Num. Images | Train | Validation | Test |
|----------|----------------|-------------|-------|------------|------|
| **VQA-CPv2** | 438K* | 121K* | 409K | 219K | - |
| **VQAv2** | 1,105K | 204K | 443K | 214K | 453K |
| **PathVQA** | 32K | 4.9K | 19K | 6K | 6K |
| **VQA-RAD** | 3.5K | 315 | 3K | - | 451 |

Table 3: Dataset indication. "*" denotes the summary in the train set only. "-" marks the not reported information. "K" stands for one thousand.

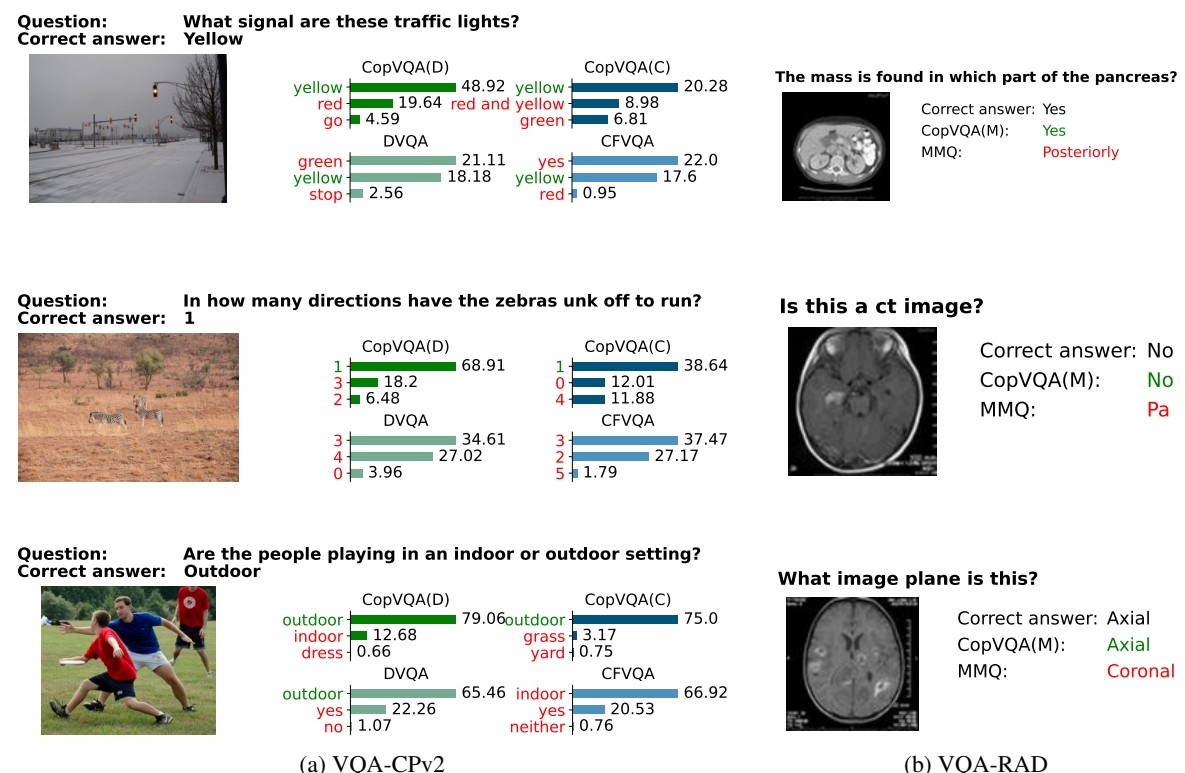

(a) VQA-CPv2
(b) VQA-RAD

Figure 10: Examples of cases effectively debiased by CopVQA over baselines. The green labels are the correct answers while the red labels are the incorrect answers.

| Test set | VQA-CPv2 | | VQAv2 | |
|----------|----------|----------|--------|----------|
| Method | Params | Duration | Params | Duration |
| CFVQA | 48.33M | x | 47.89M | x |
| +CopVQA | 47.20M | 0.96x | 46.24M | 0.91x |
| DVQA | 54.21M | x | 52.12M | x |
| +CopVQA | 51.09M | 0.87x | 51.29M | 0.93x |

| Test set | PathVQA | | VQA-RAD | |
|----------|---------|----------|----------|----------|
| Method | Params | Duration | Params | Duration |
| MMQ | 28.15M | x | 20.07M | x |
| +CopVQA | 28.06M | 0.98x | 20.02M | 0.94x |

Table 4: Comparison of model size and proportion of average training duration over 5 runs. All models are trained on the same single NVIDIA RTX A6000.

Figure 12 shows cases where all models' predictions are incorrect. It is observed that a significant portion of these cases falls into two categories: complex counting tasks and ambiguous answers. Complex counting tasks often involve intricate arrangements or a large number of objects, which pose challenges for the models in accurately counting and identifying the objects. The inherent complexity of these tasks can lead to errors in the predictions of all models, including CopVQA. Additionally, cases with ambiguous answers present difficulties for the models, as the correct answer may vary depending on the interpretation or subjective judgment. For example, the second sample can be categorized in both complex counting tasks and ambiguous answers where **(1)** the image including many people with dark clothes in a dark background, **(2)** the input is the duplication of 4

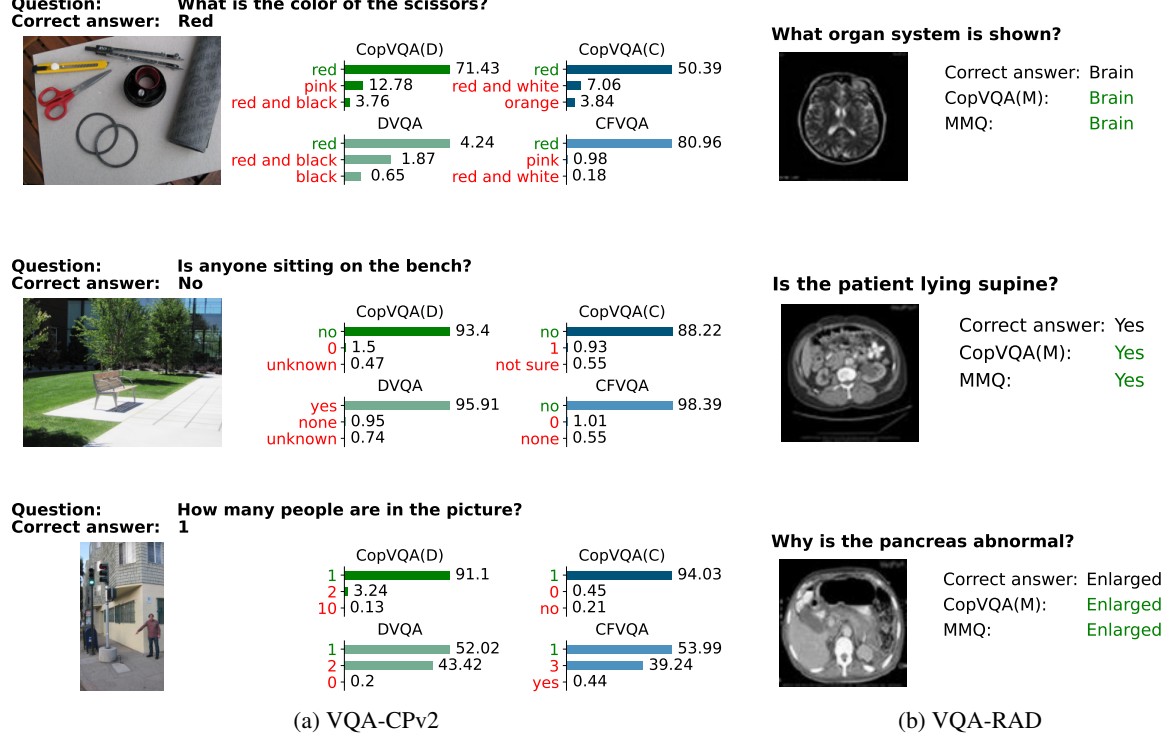

(a) VQA-CPv2            (b) VQA-RAD

Figure 11: Examples of cases where all models successfully answer the questions. The green labels are the correct answers while the red labels are the incorrect answers.

images, since **(3)** the target answer likely to only rely on 1 image piece. In such instances, all models struggle to provide accurate responses, resulting in incorrect predictions across the board. These findings emphasize the existing limitations and challenges in VQA systems when it comes to intricate counting tasks and handling ambiguous answers.

## B.2 Answers Distribution

Figure 13 indicates the analysis of answer distribution in VQA, across the train, test sets, and predictions, where all models successfully overcome the biases. Firstly, it becomes evident that biases exist in the VQA-CPv2 dataset as the answer distribution in the train and test sets significantly differ. However, despite these challenges, all models demonstrate successful debiasing. Notably, CopVQA outperforms the baseline models by producing a test set distribution that aligns more closely with the ground truth, indicating a notable improvement in its ability to generate likely answers.

Figure 14 shows cases where all models fail to perform well and biased answers persist in the predictions, several observations can be made. Firstly, there are strong biases present, indicated by the stark disparity between the answer distributions in

the train and test sets. This implies that the models have not effectively generalized from the training data to handle the biases present in the test set. Additionally, the models may struggle with providing accurate answers for categories that have limited occurrences, such as specific brand names that appear only a few times in the dataset. These challenges highlight the need for further improvement in handling biases and addressing rare answer categories to enhance the overall performance of VQA models.

## C Design principles inspired from Knowledge modularity

In the context of machine learning, modularization refers to the decomposition of complex systems or models into **smaller**, more **manageable** modules [Kahneman, 2011, Lindauer et al., 2019, Zhang et al., 2020, Kwon et al., 2019, Sayyed and Kulkarni, 2021, Garibaldi, 2021, Goyal and Bengio, 2022]. In the realm of knowledge modularity and machine learning, this principle suggests that breaking down complex tasks or models into modular components can lead to more effective and efficient learning. The advantage of this modular

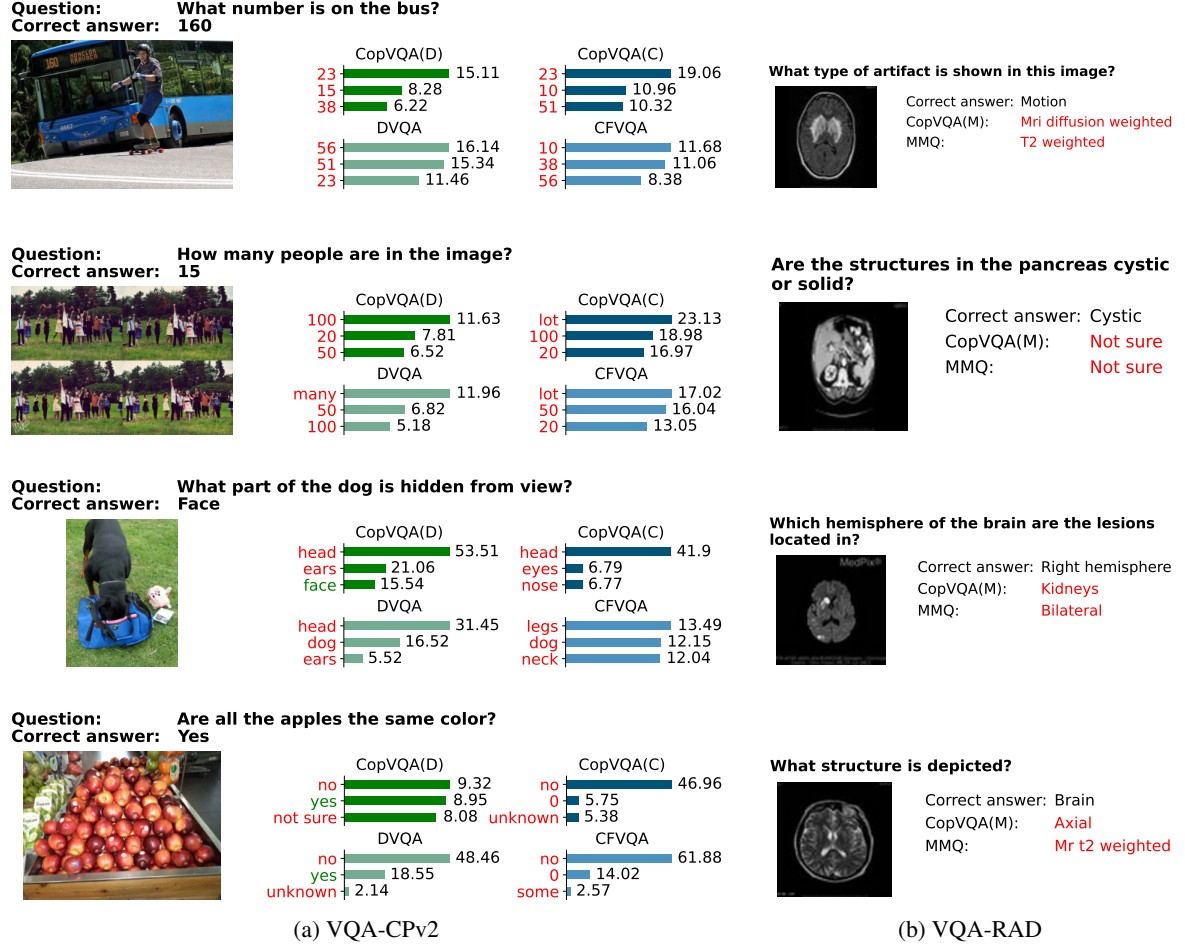

Figure 12: Examples of cases that all models failed to remove biases. The green labels are the correct answers while the red labels are the incorrect answers.

approach is that it allows for the development of specialized modules that can be individually optimized (learning independently by Gumbel-max activation function in CopVQA), leading to improved performance on specific subtasks. It also promotes reusability, as modular components can be shared or combined to address related tasks or domains. Furthermore, modularization in machine learning facilitates interpretability and explainability. Since each module focuses on a specific aspect of the task, it becomes easier to understand and analyze the contributions of each module to the overall decision-making process. This transparency can be particularly valuable in domains where interpretability is crucial, such as healthcare or autonomous systems.

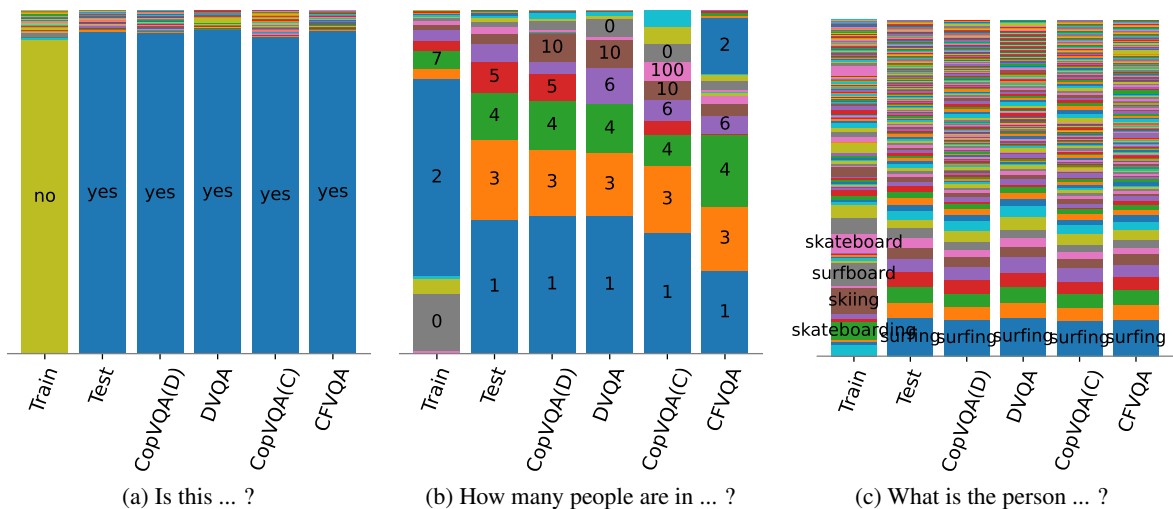

(a) Is this ... ?    (b) How many people are in ... ?    (c) What is the person ... ?

Figure 13: Answers distributions on VQA-CPv2 in the cases that all models are successful.

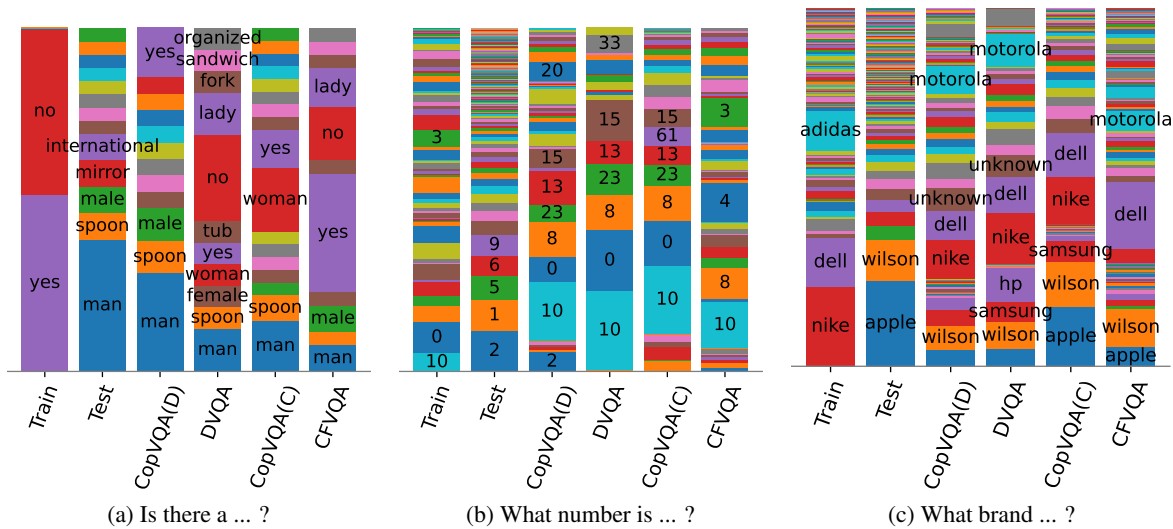

(a) Is there a ... ?    (b) What number is ... ?    (c) What brand ... ?

Figure 14: Answers distributions on VQA-CPv2 in cases where all models failed.