# OpenReview forum: "Causal Reasoning through Two Cognition Layers for Improving Generalization in Visual Question Answering"
_EMNLP/2023/Conference — EMNLP 2023 Main_

### Official Review · Reviewer_YXHL · 2023-07-31

**Soundness:** 4

**Excitement:**

4: Strong: This paper deepens the understanding of some phenomenon or lowers the barriers to an existing research direction.

**Paper Topic And Main Contributions:**

- The paper proposes CopVQA to improve OOD generation by enhancing causal reasoning that is compatible with diverse VQA baselines and domains
- This work first formulates VQA as two layers of cognitive pathways
- This paper achieve SOTA for the PathVQA dataset

**Questions For The Authors:**

Can you explain why do you add the L_M, L_U, and L_total as the final loss?

**Reasons To Accept:**

- The intuition from cognitive neuroscience about knowledge modularity and cognitive pathways is interesting
- This work first proposes a causal reasoning method, CopVQA  that formulates VQA as two layers of cognitive pathways (i.e. interpreting and answering)
- The proposed model CopVQA achieves SOTA for PathVQA dataset

**Reasons To Reject:**

I am not an expert in causality, but I try my best to read this work, the implementation part is like an increment of [1]. And for the experiment part, I am willing to see more results of other baselines on PathVQA and VQA-RAD

[1] Niu, Yulei, et al. “Counterfactual VQA: A Cause-Effect Look at Language Bias.” 2021 CVPR


**Reproducibility:**

3: Could reproduce the results with some difficulty. The settings of parameters are underspecified or subjectively determined; the training/evaluation data are not widely available.

**Reviewer Confidence:**

2: Willing to defend my evaluation, but it is fairly likely that I missed some details, didn't understand some central points, or can't be sure about the novelty of the work.

---

> ### Author Rebuttal · Authors · 2023-08-28
>
> We truly appreciate the reviewer's time and efforts in reviewing our work.
> We are glad to see `interesting`, `first proposes`, and `achieves SOTA` in your reasons to accept.
>
> ---
> > **The implementation part is like an increment of [1]. And for the experiment part, I am willing to see more results of other baselines on PathVQA and VQA-RAD**
>
>
> Thank you for your observation.
>
>
>
> First of all, we would like to clarify that our implementation does not seem `like an increment of [1]`.
>
> We provide more details  as follows:
> - We extend the main ideas (`mediator` and `subtraction of flows`) from [1].
> - However, the implementation is based on each particular baseline. That means we do not solidly improve/implement based on the implementation of [1], but  implement our method multiple times on different baselines' code base for a fair performance comparison.
>
> Regarding the baselines of PathVQA and VQA-RAD, to our best knowledge, before our submission, PathVQA has 1 baseline, which is MMQ;  and VQA-RAD has 2 baselines, which are MMQ and CLIP. We have compared our performance to these baselines in the main text.
>
> We hope this information could clarify your confusion.
>
> ---
> > **Can you explain why do you add the L_M, L_U, and L_total as the final loss?**
>
> Sure. The main reason is that we follow the ideas from [1]. The following insights would explain the loss functions design:
> - Notice that we do not include $Z_{final}$ in any loss function.  We expect each component in $Z_{final}$  to capture the desired biases.
> - By that reason, we train $Z_U$, $Z_M$, and $Z_{total}$  separately.
> - If we do not train them separately but, for instance, through a combined answer as $Z_{total}$, the individual component is not ensured to receive feedback (from the target answer) fairly and, thus, would not capture the desired biases well.
> - Although trained separately, these branches do not achieve good scores (as it is to capture biases and designed with limited capabilities) as visualized in Section 7 (Ablations study).
>
>
> We hope this information could clarify your confusion.
>
> ---
> [1] Niu, Yulei, et al. "Counterfactual VQA: A Cause-Effect Look at Language Bias." 2021 CVPR

---

### Official Review · Reviewer_ecEB · 2023-08-08

**Soundness:** 4

**Excitement:**

4: Strong: This paper deepens the understanding of some phenomenon or lowers the barriers to an existing research direction.

**Paper Topic And Main Contributions:**

This paper studies an important problem statement involving generalization in Visual Question Answering (VQA) that requires models to answer questions about images with contexts beyond the training distribution. The authors propose Cognitive pathways VQA (CopVQA) improving the multimodal predictions by emphasizing causal reasoning factors. CopVQA first operates a pool of pathways that capture diverse causal reasoning flows through interpreting} and answering stages. Next, they decompose the responsibility of each stage into distinct experts and a cognition-enabled component (CC). The two CCs strategically execute one expert for each stage at a time. Then they prioritize answer predictions governed by pathways involving both CCs while disregarding answers produced by either CC, thereby emphasizing causal reasoning and supporting generalization. They experiment on medical data and find that CopVQA improves VQA performance and generalization across baselines and domains.

**Questions For The Authors:**

See above

**Reasons To Accept:**

- Interesting and timely problem statement
- strong results on medical data
- I like cognitive pathways angle brought in to VQA
- well-written paper and interesting analysis

**Reasons To Reject:**

These are not 'reasons to reject', but some questions I have:
- I like the analysis in B.2, but was curious if authors have identified any possible data contaminations in the test set of the results they have reported in the paper?
- In NLP, Use of 'instructions' have strongly improved generalization (https://aclanthology.org/2022.acl-long.244.pdf, and https://openreview.net/forum?id=gEZrGCozdqR). Wondering authors would like to discuss if they have considering using instructions in their setup? Some of the issues authors see in sec 6.2 and 7 may be solved via instructions.
- How sensitive is the model to hyper-parameters? I see a mention in the limitation section 'Require careful fine-tuning', but was looking for more details.

**Reproducibility:**

3: Could reproduce the results with some difficulty. The settings of parameters are underspecified or subjectively determined; the training/evaluation data are not widely available.

**Reviewer Confidence:**

3: Pretty sure, but there's a chance I missed something. Although I have a good feel for this area in general, I did not carefully check the paper's details, e.g., the math, experimental design, or novelty.

---

> ### Author Rebuttal · Authors · 2023-08-28
>
> We truly appreciate the reviewer's time and efforts in reviewing our work.
> We are glad to see `interesting`, `timely`, `strong results`, and `well-written` in your reasons to accept.
>
>
>
>
>
>
>
> ---
> > **I like the analysis in B.2, but was curious if authors have identified any possible data contaminations in the test set of the results they have reported in the paper?**
>
> We appreciate your attention to this crucial aspect.
>
> In this work, we adhere to the pipeline established by prior studies, focusing on four datasets: VQA-CPv2, VQAv2, PathVQA, and VQA-RAD. It is essential to emphasize that, within the AI research community, there is no known history of reports regarding data contamination within these datasets.
>
> We hope this information clarifies your concerns.
>
>
>
> ---
> > **In NLP, Use of 'instructions' have strongly improved generalization.** Wondering authors would like to discuss if they have considered using instructions in their setup? Some of the issues authors see in sec 6.2 and 7 may be solved via instructions.
>
> Thank you for your interesting suggestion.
>
> First of all, we would like to clarify that we interpret the "Instructions" as the "Instructions Tuning" [1] technique.
>
> Next, after careful consideration of the concept of "Instructions Tuning", we would like to offer the following insights:
> - We are thrilled to observe that the prevalent issue of generalization is receiving substantial attention within the research community.
> - It is important to note that our methodology does **not** incorporate LLMs,  domain generalization, and especially explicit task definitions. Consequently, we do not immediately perceive a strong correlation between our approach and the concept of "instructions".
> - On an opposite note, we think that integrating the Cop concept into "Instructions" training could yield promising benefits based on the sets of experts and the reasoning flows. This integration might enhance generative capabilities within a specific task and facilitate the reusable modules (experts) across various tasks.
>
>
>
>
> Finally, we would really appreciate if the reviewer could help us figure out possible approaches in case we missed some angles.
>
> ---
> > **How sensitive is the model to hyper-parameters?** I see a mention in the limitation section 'Require careful fine-tuning', but was looking for more details.
>
> Thank you for your question.
>
> In the paper, we provide the analyses for observing the hyper-parameter sensitivity as our best efforts. We would like to list them as follows:
>
> - Section 7 - Figure 7: Number of experts in each MoE module.
> - Section A.3 - Figure 9: Number of experts to be activated.
>
> We would really appreciate the reviewer suggesting other analyses in case we missed some angles.
>
> [1] Instruction Tuning with FLAN - Finetuned Language Models are Zero-Shot Learners - Google Research

---

### Official Review · Reviewer_1oDT · 2023-08-11

**Soundness:** 3

**Excitement:**

3: Ambivalent: It has merits (e.g., it reports state-of-the-art results, the idea is nice), but there are key weaknesses (e.g., it describes incremental work), and it can significantly benefit from another round of revision. However, I won't object to accepting it if my co-reviewers champion it.

**Paper Topic And Main Contributions:**

The paper proposes Cognitive pathways VQA (CopVQA) to boost the causal reasoning in VQA to enhance the OOD generalization.  The paper validates the model's effectiveness on several VQA datasets.

**Questions For The Authors:**

1. Could you provide the bias distribution of the VQA dataset and specify which biases are addressed, thereby effectively validating the model's effectiveness?
2. The experiments are not detailed enough, including experimental settings, model parameters, training time, inference time, etc.

**Reasons To Accept:**

1. The idea of CopVQA is interesting，which formulates VQA as two layers of cognitive pathway to  boost the causal reasoning.
2. CopVQA achieves the better performance on several VQA datasets.

**Reasons To Reject:**

1. There is doubt about the effectiveness of the complex reasoning module, as the improvement in downstream tasks by the model is limited.
2. Compared to sota pre-trained methods, there is a significant performance gap with this method. Does it have scaling and generalization ability?
3. The method section is too redundant, and the experiments are not detailed enough, including experimental settings, model parameters, training time, inference time, etc.

**Reproducibility:**

2: Would be hard pressed to reproduce the results. The contribution depends on data that are simply not available outside the author's institution or consortium; not enough details are provided.

**Reviewer Confidence:**

4: Quite sure. I tried to check the important points carefully. It's unlikely, though conceivable, that I missed something that should affect my ratings.

---

> ### Author Rebuttal · Authors · 2023-08-28
>
> We truly appreciate the reviewer's time and efforts in reviewing our work.
> We are glad to see `interesting` and `better performance` in your reasons to accept.
>
>
>
>
>
> ---
> > **Compared to sota pre-trained methods, there is a significant performance gap with this method. Does it have scaling and generalization ability?**
>
>
>
> Thank you for your question.
>
> We acknowledge the gap between our method and the SOTA baselines. However, in these comparisons, our proposed method takes much fewer parameters, no extra data and human involvement, and less training time. In particular, CopVQA(D) requires just 2 hours of training with a single GPU, whereas MMBS demands 2 days of computation across 2 GPUs. Remarkably, CopVQA(D) achieves an Overall accuracy score on VQA-CPv2 that is only 0.49 points lower than the state-of-the-art MMBS model.
>
> Several vital characteristics substantiate our method's capacity for scaling and generalization:
> - Our approach avoids the introduction of additional parameters than the number of parameters of baselines.
> - It can represent the model's selection, including expert assignment explicitly.
> - It exhibits compatibility with a diverse range of baseline models.
>
> Consequently, our method exhibits the potential for scalability to larger datasets and more complex problems while demonstrating generalizability by incorporating causal reasoning considerations.
>
> Finally, we acknowledge room for future work to employ CopVQA on pre-trained baselines to improve the performance further.
>
>
> ---
> > **There is doubt about the effectiveness of the complex reasoning module, as the improvement in downstream tasks by the model is limited.**
>
> Thank you for your discussion.
>
>
>
>
>
>
> If we understand correctly, this review is about the `reasoning modules` and `improvement` in the general AI research. We would like to share our opinions as follows:
> - Many of the recent works improve the downstream tasks significantly by the involvement of reasoning modules, such as in dynamic modeling tasks [1,2], biology-related tasks [3,4,5], or in NLP [6,7,8,9].
> - On the other hand, the downstream task improvements may not be the ultimate goal of complex reasoning modules. They can also enhance models' transferability/reusability [1,2] across tasks, making AI systems more versatile. Therefore, facilitating causality research is a needed step.
>
>
> These points strongly drive our motivations to propose CopVQA further to facilitate the generalization and explainability via causal reasoning.
>
> The effectiveness of complex reasoning modules is a topic of ongoing debate within the AI community. We would be happy to discuss further on this point with the reviewer in case we missed some angles.
>
>
>
> ---
> > **The method section is too redundant, and the experiments are not detailed enough, including experimental settings, model parameters, training time, inference time, etc.**
>
> Thank you for your comment and question.
>
> In terms of `The method section is too redundant`, we would greatly value the reviewer's insights into specific areas that appear redundant. Any detailed feedback on this matter would greatly assist us in enhancing the clarity and conciseness of our method section.
>
> In terms of `the experiments are not detailed enough`, we have provided information related to the training details in the paper. Particularly, the number of parameters, GPU model,  and training time comparisons are in Table 4 in the Appendix and referred to in Section 6.1.
>
> Additionally, in the table below, we indicate training **hours** on a single NVIDIA RTX A6000 GPU (same as reported in the paper).
> |  |  CopVQA(D) | DVQA | CopVQA(C) | CFVQA | CopVQA(M)  | MMQ |
> |----------|----------|----------|----------|----------|---------|---------|
> | VQA-CPv2 / VQAv2 |  2.1 / 1.9 | 2.3 / 2.0 | 1.6 / 1.4 | 1.9 / 1.7 |  - | - |
> | PathVQA / VQA-RAD | - | - | - | - | 1.6 / 0.25 | 1.7 / 0.26
>
>
>
>
>
>
> For more insights on the experiments, we have provided the analyses for observing the hyper-parameter sensitivity as our best efforts in the paper. We would like to list them as follows:
>
> - Section 7 - Figure 7: Number of experts in each MoE module.
> - Section A.3 - Figure 9: Number of experts to be activated.
>
>
> We hope this information clarifies your concern and helps for your assessment.
>
> ---
> > **Could you provide the bias distribution of the VQA dataset and specify which biases are addressed, thereby effectively validating the model's effectiveness?**
>
> Sure. It is important to note that we are not allowed to provide the answer distribution as Figure 4 from the paper during this rebuttal phase due to constraints on attaching figures or links. Nevertheless, the explanation and analysis below comprehensively fulfill this purpose.
>
> First of all, we would like to clarify `which biases are addressed`. The biases that need to be overcome in the VQAv2 dataset (as well as the VQA task) are some answers account for a large proportion compared to other answers' proportions within a question category. The examples in the following part will further visualize the biased answers in the VQAv2 dataset.
>
>
>
> Next, regarding `Could you provide the bias distribution?`, in the following tables, we provide the indications of the number of instances and correctness in the format `# / Acc` in different questions and answers that contain biases. The `Acc` evaluates the accuracy of these `#` answers,  which would clarify your concerns about, for instance, exactly `which biases are addressed` to `validate the model's effectiveness`.
>
>
>
>
>
>
>
>
> Let's analyze this **first example** for questions starting with "**Is this**". Please notice that the VQAv2  is biased but not an OOD dataset, as its biases are similar in the train and test sets.
> -  `yes` accounts for 46%,  `no` accounts for 47%, and `inside` takes 0.19% in the training set.
> -  `yes` accounts for 45%,  `no` accounts for 46%, and `inside` takes 0.19% in the test set.
>
>
> In this case, `yes` and `no` are biased answers. We want to test if CopVQA can overcome them and still predict the answer `inside`.
> - As illustrated in the table below, it is evident that CopVQA can predict low-frequency answers as `inside` with high accuracy,  while the baselines tend to overlook these less common cases.
>
> | `Is this` ... | Train | Test | CopVQA(D) | DVQA | CopVQA(C) | CFVQA |
> |----------|----------|----------|----------|----------|----------|----------|
> | **# / Acc** for `yes` | 7502/- | 3529/- | 3591 / 0.90 | 3623/ 0.82 | 3525 / 0.86 | 3741 / 0.79 |
> | **# / Acc** for `no` | 7656/- | 3620/- | 3632 / 0.89 | 3773/ 0.80 | 3597 / 0.84 | 3963 / 0.72 |
> | **# / Acc** for `inside` | 32/- | 15/- | 12 / 1.0 | 0 / NaN | 4 / 1.0 | 0 / NaN |
>
> Moving to the **second example**, with questions starting with "**Which**":
> -  `left` accounts for 16%,  `right` accounts for 18%, and `baseball` takes 1.4% in the training set.
> -  `left` accounts for 15%,  `right` accounts for 16%, and `baseball` takes 1.1% in the test set.
>
> In this scenario,  `left`  and  `right`  are examples of biased answers. We aim to test whether CopVQA can successfully address this bias and still predict the answer  `baseball`.
> - As demonstrated in the table below, it is clear that CopVQA can generate low-frequency answers as `baseball` with high accuracy, while the baseline models often fail to capture these less common cases.
>
>
> | `Which` ... | Train | Test | CopVQA(D) | DVQA | CopVQA(C) | CFVQA |
> |----------|----------|----------|----------|----------|----------|----------|
> | **# / Acc** for `left` | 803/- | 385/- | 392 / 0.84 | 443 / 0.32 | 392 / 0.76 | 594 / 0.29 |
> | **# / Acc** for `right` | 859/- | 396/- | 399 / 0.81 | 497/ 0.41 | 390 / 0.74 | 630 / 0.42 |
> | **# / Acc** for `baseball` | 67/- | 28/- | 24 / 0.95 | 1 / 1.0 | 19 / 0.94 | 0 / NaN |
>
>
> To conclude, these analyses demonstrate the ability of CopVQA to overcome biased answers.
>
>
> We hope this information and analysis clarifies your question.
>
>
>
> ---
> [1] Neural Production Systems: Learning Rule-Governed Visual Dynamics - Anirudh Goyal, Aniket Didolkar, Nan Rosemary Ke, Charles Blundell, Philippe Beaudoin, Nicolas Heess, Michael Mozer, Yoshua Bengio
>
> [2] Reusable Slotwise Mechanisms - Trang Nguyen, Amin Mansouri, Kanika Madan, Khuong Nguyen, Kartik Ahuja, Dianbo Liu, Yoshua Bengio
>
> [3] Modeling in systems biology: Causal understanding before prediction? - Szilvia Barsi, Bence Szalai
>
>
> [4] DynGFN: Towards Bayesian Inference of Gene Regulatory Networks with GFlowNets - Lazar Atanackovic, Alexander Tong, Jason Hartford, Leo J. Lee, Bo Wang, Yoshua Bengio
>
> [5] Causal Concepts in Biology: How Pathways Differ from Mechanisms and Why It Matters - Lauren N. Ross
>
> [6] Measuring semantic similarity of clinical trial outcomes using deep pre-trained language representations - Anna Koroleva, Sanjay Kamath, Patrick Paroubek
>
> [7] Causal curiosity: Rl agents discovering self-supervised experiments for causal representation learning - Sumedh A Sontakke, Arash Mehrjou, Laurent Itti, Bernhard Schölkopf
>
> [8]  Causal Mediation Analysis for Interpreting Neural NLP: The Case of Gender Bias - Jesse Vig, Sebastian Gehrmann, Yonatan Belinkov, Sharon Qian, Daniel Nevo, Simas Sakenis, Jason Huang, Yaron Singer, Stuart Shieber
>
> [9] Counterfactual Generative Networks - Axel Sauer, Andreas Geiger

---

### Official Review · Reviewer_UGPV · 2023-08-12

**Soundness:** 4

**Excitement:**

4: Strong: This paper deepens the understanding of some phenomenon or lowers the barriers to an existing research direction.

**Paper Topic And Main Contributions:**

This paper tried to improve the multimodal predictions in Visual Question Answering by emphasizing causal reasoning factors through a improved causal graph. The scope of this paper particularly focuses on improving OOD generalizability of current VQA models.

The main contributions of this paper are two-fold:
1. Proposed a VQA model that improves both OOD generalization and benchmark performance.
2. Enhanced causal reasoning factors in VQA with two-layer reasoning pathway, which shows novelty over previous causal VQA researches.

**Questions For The Authors:**

a. Some details of such model remains unclear: What kind of MoE models are involved in those two versions of CopVQA?
b. In 6.2 "Discussion on the debiased samples", it claims to reduce biases from the texts in question. However, there is no statistics of such phenomenon: How many instances are there in which DVQA/CFVQA/MMQ is misled by "Is this" and how many of them are resolved by CopVQA.
c. The training details are missing: How long does it take to train CopVQA on what kind of machine?

**Reasons To Accept:**

a. The quantitative results of the newly proposed framework indicate a new SOTA.
b. Newly designed cognitive pathways are innovative and the corresponding loss function described such reasoning paths well.
c. The qualitative analyses proves the effectiveness of such two-layer reasoning path way.
d. Enough case studies explains the advantages of such method.
e. Paper is well-organized.

**Reasons To Reject:**

a. Some details of such model remains unclear.
b. In 6.2 "Discussion on the debiased samples", it claims to reduce biases from the texts in question. However, there is no statistics of such phenomenon.
c. The training details about training time and hardware requirements are missing.

**Reproducibility:**

4: Could mostly reproduce the results, but there may be some variation because of sample variance or minor variations in their interpretation of the protocol or method.

**Reviewer Confidence:**

3: Pretty sure, but there's a chance I missed something. Although I have a good feel for this area in general, I did not carefully check the paper's details, e.g., the math, experimental design, or novelty.

---

> ### Author Rebuttal · Authors · 2023-08-28
>
> We truly appreciate the reviewer's time and efforts in reviewing our work.
> We are glad to see `innovative`, `effectiveness`, `enough case studies`, and `well-organized` in your reasons to accept.
>
>
>
> ---
> > a. **Some details of such model remain unclear: What kind of MoE models are involved in those two versions of CopVQA?**
>
> Thank you for your questions, and we are so sorry for the confusion on the MoE design.
>
> The details of the MoE designs are described in Section 4.3 of the paper. We would also highlight them as well as provide some more information as follows:
>
>
>
> - **Gating Model**: Hard Gating. We design the gating model as an MLP followed by a Gumbel-softmax with hard selection to select  one of the experts to process the given input. We use the Straight-through trick to train this gating model.
> - **Experts with Separate Parameters**: Each expert carries distinct parameters, which are randomly initialized.
> - We tune the **number of experts** for $\mathcal{M}_I$ and $\mathcal{M}_A$ for each dataset and baseline and provide the tuning comparison in Section 7.
> - In our main design, the gating model only **activates 1 expert** at a time. We have also analyzed the performance when activating multiple experts and presented the results in Section A.3. The analysis shows that activating one expert is always better, proving the importance of disentanglement in our design.
> - We implement the MoE by ourselves. The MoE architecture design strategy is the same for $\mathcal{M}_I$ and $\mathcal{M}_A$.
>
> We hope the information above could clarify your concerns.
>
> ---
> > **b. In 6.2 "Discussion on the debiased samples", it claims to reduce biases from the texts in question. However, there are no statistics of such phenomenon: How many instances are there in which DVQA/CFVQA/MMQ is misled by "Is this?" and how many of them are resolved by CopVQA.**
>
> Thank you for your insightful comment.
>
> The following tables indicate the number of instances and correctness in the format `# / Acc` in different questions and answers. The `Acc` evaluates the accuracy of these `#` answers,  which would clarify your concerns about, for instance, `how many of them are resolved by CopVQA`.
>
> According to the tables, we can see that CopVQA(D) is the model that solves most biased cases within a question category as it acquires the closest `#` compared to ones of the test set and high `Acc`, followed by CopVQA(C).
>
>
> | `Is this` ... | Train | Test | CopVQA(D) | DVQA | CopVQA(C) | CFVQA |
> |----------|----------|----------|----------|----------|----------|----------|
> | **# / Acc** for `yes` | 0/- | 11031/- | 10991 / 0.99 | 11084 / 0.92 | 10866/ 0.99 | 11071 / 0.91 |
> | **# / Acc** for `no` | 11276/- | 0/- | 1 / 0.0 | 165 / 0.0 | 71 / 0.0 | 46 / 0.0 |
>
>
> | `How many` ... | Train | Test | CopVQA(D) | DVQA | CopVQA(C) | CFVQA |
> |----------|----------|----------|----------|----------|----------|----------|
> | **#** / **Acc** for `2` | 15270/- | 0/- | 492 / 0.0 | 2731 / 0.0 | 655 / 0.0  | 13406 / 0.0 |
> | **# / Acc** for `1` | 0/- | 15768/- | 15590 / 0.97 | 13894 / 0.76 | 14577 / 0.91 | 1855 / 0.47 |
>
>
>
>
>
>
> ---
> > **c. The training details are missing: How long does it take to train CopVQA on what kind of machine?**
>
> Thank you for your question.
>
>
>
>
> In the paper, we have provided information related to the training details. Particularly, the number of parameters, GPU model, and training time comparisons are in Table 4 in the Appendix and referred to in Section 6.1.
>
>
> In addition, in the table below, we indicate training **hours** on a single NVIDIA RTX A6000 GPU (same as reported in the paper).
> |  |  CopVQA(D) | DVQA | CopVQA(C) | CFVQA | CopVQA(M)  | MMQ |
> |----------|----------|----------|----------|----------|---------|---------|
> | VQA-CPv2 / VQAv2 |  2.1 / 1.9 | 2.3 / 2.0 | 1.6 / 1.4 | 1.9 / 1.7 |  - | - |
> | PathVQA / VQA-RAD | - | - | - | - | 1.6 / 0.25 | 1.7 / 0.26
>
>
> We hope the information above could help for your assessment.

---

### Meta-Review · Area_Chair_FRBf · 2023-09-23

**Recommendation:** 5

**Metareview:**

The paper aims to enhance out-of-distribution (OOD) generalization in Visual Question Answering (VQA) by emphasizing causal reasoning through a novel construct, the Cognitive pathways VQA (CopVQA). This model promotes multimodal predictions and primarily works in two layers of cognitive pathways. The paper has garnered attention for its state-of-the-art (SOTA) performance on the PathVQA dataset and its innovative approach to introducing cognitive pathways into VQA. The novel two-layer reasoning pathway provides a fresh perspective on causal VQA research, especially among tons of LLM-related papers. This  work also provides a comprehensive qualitative analysis supporting the effectiveness of the two-layer reasoning pathway. Reviewers unanimously rated this work positively, but authors are encouraged to further improve the paper based on the suggestions given in the reviews.

---

### Decision · Program_Chairs · 2023-10-07

**Decision:**

Accept-Main

**Comment:**

The paper aims to enhance out-of-distribution (OOD) generalization in Visual Question Answering (VQA) by emphasizing causal reasoning through a novel construct, the Cognitive pathways VQA (CopVQA). This model promotes multimodal predictions and primarily works in two layers of cognitive pathways. The paper has garnered attention for its state-of-the-art (SOTA) performance on the PathVQA dataset and its innovative approach to introducing cognitive pathways into VQA. The novel two-layer reasoning pathway provides a fresh perspective on causal VQA research, especially among tons of LLM-related papers. This  work also provides a comprehensive qualitative analysis supporting the effectiveness of the two-layer reasoning pathway. Reviewers unanimously rated this work positively, but authors are encouraged to further improve the paper based on the suggestions given in the reviews.